# ProDiff: Prototype-Guided Diffusion for Minimal Information Trajectory Imputation

**Tianci Bu** [* 1]  **Le Zhou** [* 1]  **Wenchuan Yang** [* 1]  **Jianhong Mou** [1]  **Kang Yang** [2]  **Suoyi Tan** [1]  **Feng Yao** [1]
**Jingyuan Wang** [3 4 5]  **Xin Lu** [1]

## Abstract

Trajectory data is crucial for various applications but often suffers from incompleteness due to device limitations and diverse collection scenarios. Existing imputation methods rely on sparse trajectory or travel information, such as velocity, to infer missing points. However, these approaches assume that sparse trajectories retain essential behavioral patterns, which place significant demands on data acquisition and overlook the potential of large-scale human trajectory embeddings. To address this, we propose ProDiff, a trajectory imputation framework that uses only two endpoints as minimal information. It integrates prototype learning to embed human movement patterns and a denoising diffusion probabilistic model for robust spatiotemporal reconstruction. Joint training with a tailored loss function ensures effective imputation. ProDiff outperforms state-of-the-art methods, improving accuracy by 6.28% on FourSquare and 2.52% on WuXi. Further analysis shows a 0.927 correlation between generated and real trajectories, demonstrating the effectiveness of our approach.

## 1. Introduction

Mining spatio-temporal patterns from trajectory data has broad applications, such as infectious diseases control, human behavioral analysis, and urban planning (Jia et al., 2020;

---

[*]Equal contribution [1]College of Systems Engineering, National University of Defense Technology, Changsha, China [2]School of Information, Renmin University of China, Beijing, China [3]School of Computer Science and Engineering, Beihang University, Beijing, China [4]School of Economics and Management, Beihang University, Beijing 100191, China [5]Engineering Research Center of Advanced Computer Application Technology, Ministry of Education. Correspondence to: Jingyuan Wang <jywang@buaa.edu.cn>, Xin Lu <xin.lu.lab@outlook.com>.

*Proceedings of the 42nd International Conference on Machine Learning*, Vancouver, Canada. PMLR 267, 2025. Copyright 2025 by the author(s).

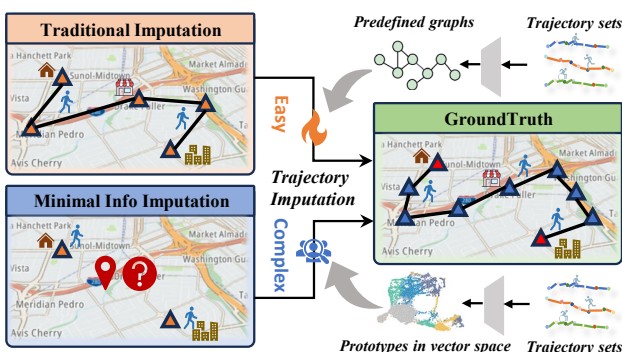

*Figure 1.* Comparison of traditional and proposed trajectory imputation. Traditional methods preserve movement patterns but impose device constraints and rely on predefined graphs. Our approach directly embeds trajectories into vector space for minimal information imputation.

Zhang et al., 2023; Zheng et al., 2014; Hettige et al., 2024; Ji et al., 2022b). Such data primarily originates from Location-Based Services (LBS) using cell tower signals (Lu et al., 2012), satellite-based systems such as GPS, GLONASS, BeiDou, QZSS, and Galileo, as well as IP-based location methods utilized by online platforms.

Most of the trajectory mining tasks (Bao et al., 2021; Yao et al., 2017; Wang et al., 2018) and methods(Li et al., 2018; Huang et al., 2022; Shen et al., 2020) are based on the assumption of complete and accurate trajectory data (Chen et al., 2024), making them sensitive to the granularity and accuracy of sampled data. However, contemporary location data collection, reliant on mobile networks or satellite communications, is often hindered by base station coverage gaps, signal instability, and environmental interference, leading to frequent missing data. Traditional methods like linear interpolation (Blu et al., 2004) and vector autoregressive models (Lütkepohl, 2013) provide efficient solutions but often fail to capture the full data distribution. Deep learning-based imputation methods like Wu et al. (2023); Du et al. (2023); Xia et al. (2021) capture spatial-temporal dependencies using self-attention mechanisms or convolutional neural network, while some methods based on graph

neural networks (Chen et al., 2023; Wei et al., 2024) rely on predefined graph structures to extract spatial-temporal features. Recently, generative methods such as generative adversarial networks (GANs) (Jiang et al., 2023b) and variational autoencoders (VAEs) (Chen et al., 2021) have shown promise in trajectory synthesis, and the emergence of denoising diffusion probabilistic models (DDPMs) (Ho et al., 2020) has further advanced the field. For instance, DiffTraj (Zhu et al., 2024a) leverages diffusion model to capture group-level trajectories, generating synthetic trajectory data while preserving privacy.

Despite progress, existing trajectory imputation methods face notable limitations. First, they typically assume that sampled trajectories, despite large intervals, retain essential movement patterns, interpolating local points using global trajectories thereby imposing constraints on devices and operational environments. Second, they fail to fully embed the vast amount of unlabeled human trajectories, which exhibit consistent macro-level patterns that enable imputation under more relaxed conditions, as shown in Fig. 1. Although recent work (Wei et al., 2024) has leveraged unlabeled trajectories to aid imputation, it required the graph structures as the foundamental element for prediction.

To address these limitations, we introduce ProDiff, a framework integrating Prototype learning with a denoising Diffusion probabilistic model. ProDiff operates under minimal information constraints, modeling a trajectory as a sequence of points and interpolating missing locations using only two endpoints within a fixed-length window. This relaxes the prior assumption that sparse trajectories must retain essential movement patterns. ProDiff consists of two key components: (i) Diffusion-based generative model: The diffusion model reconstructs human movement by iteratively denoising from a latent space, offering reliable spatiotemporal modeling. (ii) Prototype-based condition extractor: This module learns prototypes that represent individual movement patterns, embedding diverse trajectories into a vector space through self-supervised learning. Given known trajectory information as queries, it extracts a comprehensive pattern representation to guide the diffusion model in generating realistic, individualized trajectories. To effectively couple these two components, we design a joint training loss function that integrates generative and prototype learning objectives. This ensures a more compact embedding space while mitigating independent error accumulation and irreversible information loss typically introduced by multi-stage training. Experimental results demonstrate the effectiveness of the proposed ProDiff and prove that the captured underlying trajectory structures can signficantly improve the imputation accuracy.

In summary, the contributions of this work are as follows:

- We relax the prior assumption that sparse trajectories inherently retain movement patterns and introduce trajectory imputation under minimal information constraints.

- We propose a prototype-based condition extractor that embeds human trajectories into a vector space, capturing macro-level behavioral patterns for the first time in trajectory imputation.

- We develop ProDiff, a framework that jointly optimizes generative modeling and prototype learning, effectively reconstructing missing trajectory data while reducing independent errors.

- We conduct extensive experiments on WuXi (Song et al., 2017) and Foursquare (Yang et al., 2014), demonstrating superior imputation accuracy across different trajectory window sizes. Our code is available at `https://github.com/b010001y/ProDiff`.

## 2. Related Work

Please refer to Appendix A for an extensive discussion of related work. Here we provide its summary.

**Spatial-Temporal Sequence Imputation.** Traditional imputation methods evolved from simple statistical approaches like linear interpolation (Blu et al., 2004) to probabilistic frameworks such as PCA and Bayesian networks (Qu et al., 2009; Shi et al., 2013), are fast but often too simple to capture complex distributions. Deep learning revolutionized the field through two paradigms: non-generative models like GRU-D (Che et al., 2018) with temporal decay mechanisms and SAITS (Du et al., 2023) using masked self-attention, and generative approaches where diffusion models like Diffusion-TS (Yuan & Qiao, 2024) now dominate by disentangling trend-seasonality components.

**Trajectory Data Mining.** Trajectory analysis spans forecasting, estimation, and anomaly detection. CNN/RNN architectures (Bao et al., 2021; Yang et al., 2017) pioneered point-wise prediction, while road-aware models like WDR (Wang et al., 2018) advanced travel time estimation through road network embeddings. Anomaly detection evolved from RNN-based classifiers (Song et al., 2018) to latent space methods like GM-VSAE (Liu et al., 2020). The emerging mobility generation field bridges forecasting and synthesis, exemplified by DiffTraj (Zhu et al., 2024a) applying raw GPS diffusion, yet lacks physics-aware trajectory topology preservation – a gap our work addresses.

**Mobility Data Synthesizing.** Early synthesis relied on statistical approximations (Simini et al., 2021) until VAEs (Chen et al., 2021) and GANs (Jiang et al., 2023b) introduced deep generative modeling. Graph-based innovations like RNTrajRec (Chen et al., 2023) captured semantics

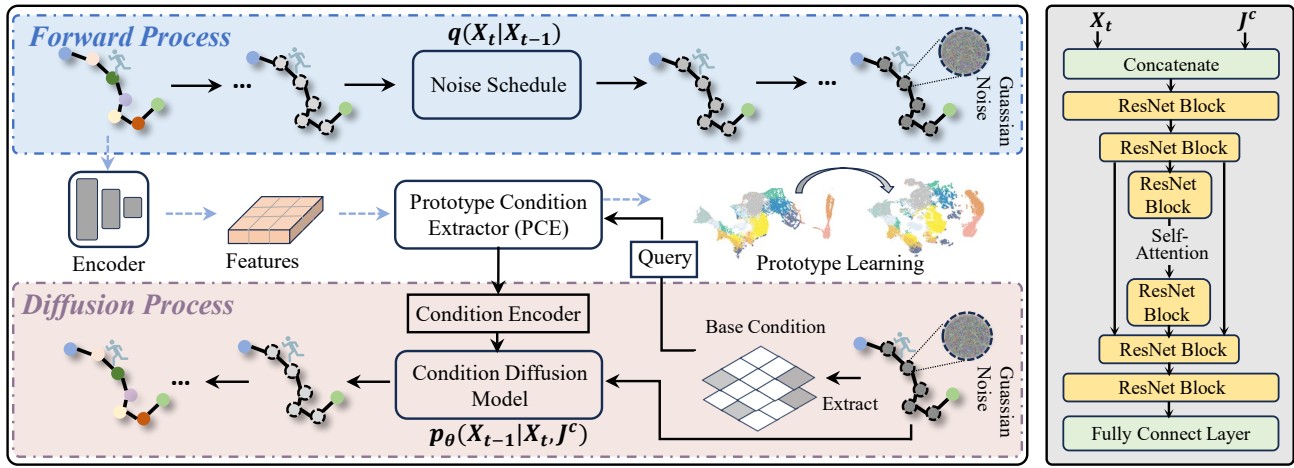

*Figure 2.* Left illustrates how prototype learning and diffusion models interact. The diffusion process progressively corrupts trajectories with Gaussian noise, preserving only the endpoints, while prototype learning embeds trajectories and extracts patterns. During denoising, prototype-based conditions, combined with endpoint features, guide the diffusion model. A joint loss function optimizes both components, ensuring effective trajectory reconstruction. Right is the architecture of the diffusion base model.

through spatial-temporal transformers, while attention architectures (Xia et al., 2021) explicitly modeled cross-region dependencies. Modern diffusion frameworks such as ControlTraj (Zhu et al., 2024b) enable conditional generation via traffic signal conditioning but remain resolution-rigid.

## 3. ProDiff Model

### 3.1. Problem Definition

**Definition 3.1. Spatio-Temporal Trajectory.** A spatio-temporal trajectory is a sequence of human activity points, denoted as $\mathbf{x}_{i,j} \in \mathbb{R}^n$, where $n$ represents the number of attributes. Each point consists of *time*, *longitude*, and *latitude*, *i.e.*, $\mathbf{x}_{i,j} = \{t_{i,j}, lon_{i,j}, lat_{i,j}\}$, satisfying $t_{i,j} < t_{i,j+1}$. The trajectory of an individual $i$ is defined as $\mathbf{X}_i = [\mathbf{x}_{i,1}, ..., \mathbf{x}_{i,l}]$, where $l$ is the trajectory length.

**Definition 3.2. Trajectory Sequence Window.** To process trajectories, we define a sliding window of size $k$ $(k < l)$ that partitions a trajectory $\mathbf{X}_i$ into overlapping segments. Each segment is represented as $\mathbf{S}_p = [\mathbf{s}_{p,1}, ..., \mathbf{s}_{p,k}]$, yielding $l - k + 1$ segments per trajectory. Given $M$ trajectories, the total number of segments for a fixed $k$ is $\sum_i^M (l_i - k + 1)$, where trajectories shorter than $k$ are discarded.

**Definition 3.3. Minimal-Information Imputation.** Given a trajectory $\mathbf{X}_i = [\mathbf{x}_{i,1}, ..., \mathbf{x}_{i,l}]$, where each point $\mathbf{x}_{i,j} = \{t_{i,j}, lon_{i,j}, lat_{i,j}\}$ represents a spatio-temporal coordinate, we define the *minimal-information imputation problem* as reconstructing $\mathbf{x}_{i,2}, ..., \mathbf{x}_{i,l-1}$ given only the endpoints $\mathbf{x}_{i,1}$ and $\mathbf{x}_{i,l}$.

### 3.2. Base Network Components

**Diffusion Base Model.** To capture spatiotemporal dependencies in trajectory imputation, we employ a 1D-UNet with residual network (ResNet) blocks. The 1D-UNet consists of down-sampling and up-sampling modules, linked by a self-attention layer. Each module encodes hidden features using group normalization, nonlinear activation, and 1D-CNN layers. The self-attention mechanism refines trajectory representations via:

$$\text{Self-Attn}(Q_h, K_h, V_h) = \text{Softmax}\left(\frac{Q_h K_h^T}{\sqrt{d_h}}\right) V_h, \quad (1)$$

where $Q_h$, $K_h$, and $V_h$ are derived from hidden features $\mathbf{h}$. The features obtained through self-attention are then passed through the up-sampling module to output the predicted noise, as shown in the right of Fig. 2.

**Base Condition.** Trajectory imputation relies on reconstructing intermediate points from the trajectory endpoints. Specifically, given a set of trajectory points $\mathbf{S}_i = [\mathbf{s}_{i,1}, ..., \mathbf{s}_{i,k}] \in \mathbb{R}^{k \times d}$, where $k$ denotes the trajectory length, we generate a mask $\mathbf{M} = [\mathbf{m}_1, ..., \mathbf{m}_k] \in \mathbb{R}^k$ which is corresponding to $\mathbf{S}_i$. For any element $\mathbf{m}_j$:

$$\mathbf{m}_j = \begin{cases} 1, & \text{if } j = 0 \text{ or } j = k, \\ 0, & \text{otherwise.} \end{cases} \quad (2)$$

This mask, when applied to trajectory points, encodes the locations to acquire the base condition $\mathbf{B}^c$ while guiding the diffusion model in reconstructing the missing points.

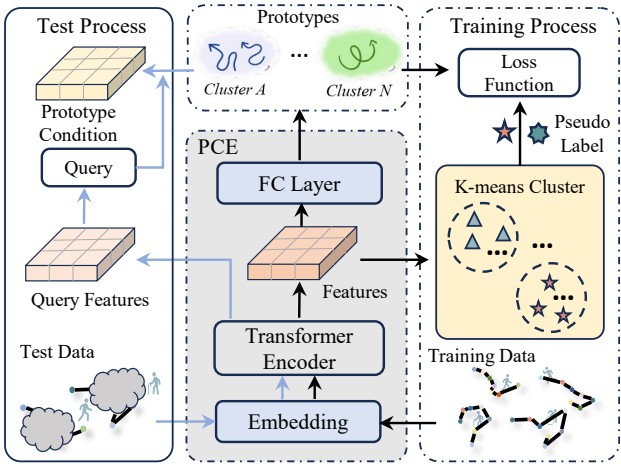

*Figure 3.* Composition of prototype condition extractor and its workflow during the training and test (black and blue lines).

### 3.3. Prototype Condition Extractor

**Embedding Trajectory Data.** To exploit large-scale un-labeled data, we introduce a Prototype Condition Extractor (PCE) that embeds trajectories into vector space and extracts latent movement patterns. For each trajectory $\mathbf{S}_i = [\mathbf{s}_{i,1}, ..., \mathbf{s}_{i,k}] \in \mathbb{R}^{k \times d}$ of window size $k$, the trajectory representation $\mathbf{H}_i$ is computed as:

$$\mathbf{H}_i = \sum_{j}^{k} \left( \text{Encoder}(\mathbf{s}_{i,j}) \right). \tag{3}$$

Prototypes $\mathbf{P} \in \mathbb{R}^{N_p \times d_p}$ where $N_p$ denotes the number of prototypes and $d_p$ represents the embedded dimension are then generated via a fully connected layer:

$$\mathbf{P} = \mathbf{W_p}\mathbf{H_p} + \mathbf{b_p}, \tag{4}$$

where $\mathbf{W_p}$ and $\mathbf{b_p}$ are learnable parameters. Since the hidden feature $\mathbf{H_p}$ is considered as trajectory representation which is summed up from the features of all points in trajectory $\mathbf{S}$. Moreover, prototypes $\mathbf{P}$ are generated by $\mathbf{H_p}$ expressing the generic movement pattern of trajectory $\mathbf{S}$, which are iteratively refined and serve as conditioning features for inference.

**Conditioning the Diffusion Model.** While the base condition serves as a guide, it often provides implicit information, making it difficult for the model to derive sufficient insights directly. To enhance the diffusion model's guidance, we encode trajectory data into queries $\mathbf{Q}^b = \{Q_1, ..., Q_B\} \in \mathbb{R}^{B \times d}$ of $B$ trajectories and project them into the prototype space using:

$$\mathbf{D} = \left[ \text{Dis}(\mathbf{Q}_b, P_1), ..., \text{Dis}(\mathbf{Q}_b, P_{N_p}) \right], \tag{5}$$

$$\mathbf{P}^c = \mathbf{D}^T \mathbf{P}. \tag{6}$$

Here, $\mathbf{P}^c$ represents the prototype-conditioned feature, aligning trajectory embeddings with learned movement patterns. $\text{Dis}(\mathbf{Q}^b, P_i)$ can be an arbitrary distance function between query $\mathbf{Q}^b$ and $i^{th}$ prototype $P_i$. With encoder optimization, the prototypes representing movement patterns are refined, and the PCE can effectively enhance the diffusion model's guidance by matching the base condition with the prototype and generating a comprehensive prototype condition.

To integrate the base condition and prototype condition, we encode and combine them using a Wide & Deep (WD) network, which contains two fully connected layers for each condition. Then the final joint condition $\mathcal{J}^c$ is formulated as:

$$\mathcal{J}^c = WD(\mathbf{B}^c) + WD(\mathbf{P}^c). \tag{7}$$

Fig. 3 details the specific workflow of the prototype network. On the right and middle sections, complete trajectories are used to train the prototype network, enhancing the generation of prototypes that accurately represent movement patterns. This training is optimized through unsupervised contrastive loss and the joint loss function. On the left and middle sections, during testing, trajectories are encoded and used to query the trained prototypes to generate the prototype condition.

### 3.4. Jointly Training Objective

Given i.i.d. samples $\mathbf{Z} \sim p$, a diffusion probabilistic model approximates the data distribution by learning $p_\theta(\mathbf{Z})$. In the *forward process*, Gaussian noise diffuses the data via the stochastic differential equation (SDE):

$$d\mathbf{Z} = \mathbf{f}(\mathbf{Z}, t)dt + g(t)d\mathbf{w}, \tag{8}$$

where $\mathbf{f}(\cdot)$ is the drift coefficient, $g(\cdot)$ is the diffusion coefficient, and $\mathbf{w}$ is a standard Wiener process. The reverse process, conditioned on $\mathcal{J}^c$, is given by:

$$d\mathbf{Z} = \left[ \mathbf{f}(\mathbf{Z}, t) - g(t)^2 \nabla_{\mathbf{Z}} \log p_t\left(\mathbf{Z}|\mathcal{J}^c\right) \right] dt + g(t)d\bar{\mathbf{w}}. \tag{9}$$

where $\nabla_{\mathbf{Z}} \log p_t(\mathbf{Z}|\mathcal{J}^c)$ is the conditional score function. The denoising network $\epsilon_\theta$ estimates this score function:

$$\epsilon_\theta(\mathbf{Z}_t, t, \mathcal{J}^c) \simeq -g(t)^2 \nabla_{\mathbf{Z}} \log p_t(\mathbf{Z}|\mathcal{J}^c), \tag{10}$$

where the joint condition $\mathcal{J}^c = f_\gamma(\mathbf{Z}_0)$. The joint loss function is:

$$\mathcal{L}_J(\theta, \gamma) = \mathbb{E}_{t \sim \mathcal{U}} \mathbb{E}_{\mathbf{Z}_0 \sim p, \epsilon \sim \mathcal{N}} \left[ \|\epsilon - \epsilon_\theta(\mathbf{Z}_t, t, f_\gamma(\mathbf{Z}_0))\|^2 \right]. \tag{11}$$

$\theta$ and $\gamma$ are the optimized parameters of denoising network and joint condition extraction network.

To enhance prototype learning for unsupervised trajectory data, we introduce additional loss functions to refine $f_\gamma$

and capture semantic movement patterns. The first classification consistency loss, $\mathcal{L}_{C1}$, enforces alignment between K-means clustering and prototype-based learning. Given trajectory features $\mathbf{H_P} = \{\mathbf{H}_1, \mathbf{H}_2, ...\}$ and $N_c$ clusters, K-means assigns pseudo-labels $p_{kmeans}$, which guide prototype learning via:

$$\mathcal{L}_{C1}(\gamma) = -\sum_{i=1}^{N_c} p_{kmeans}^i \log(q_{proto}^i), \quad (12)$$

where $q_{proto}^i$ represents the prototype-assigned label. To ensure a compact, discriminative feature space, we employ a contrastive loss that optimizes prototype separation. Given trajectory features $\mathbf{H_P}$ and prototypes $\mathbf{P} = \{P_1, ..., P_{N_p}\}$, let $\mathbf{P}^+$ and $\mathbf{P}^-$ be the closest and farthest prototypes, respectively. The loss is defined as:

$$\mathcal{L}_{C2}(\gamma) = \mathbb{E}\big[\max\big(0, d(\mathbf{H}_i, \mathbf{P}^+) - d(\mathbf{H}_i, \mathbf{P}^-) + m\big)\big], \quad (13)$$

where $m$ is a margin ensuring separation, and it $d(\cdot, \cdot)$ is a distance metric (e.g., Euclidean). Afterward, the final objective integrates all loss functions:

$$\mathcal{L}(\theta, \gamma) = \lambda_1 \mathcal{L}_J(\theta, \gamma) + \lambda_2 \mathcal{L}_{C1}(\gamma) + \lambda_3 \mathcal{L}_{C2}(\gamma), \quad (14)$$

where $\lambda_1, \lambda_2, \lambda_3$ control the weight of each term. The full training process is outlined in Algorithm 1.

### 3.5. Inference Processes

In the Inference process, given information about two points in a trajectory with sequential order, the corresponding base condition can be generated according to Eq. 2. The base condition is utilized as a query and projected into the space of trained robust prototypes to obtain the prototype condition, and finally the joint condition is obtained through Eq. 7. Then, the inference process conduct the trained denoising function $\epsilon_\theta$ to denoise from a standard Gaussian noise $\mathbf{Z_t}$ step by step. A more detailed algorithm can be found in Algorithm 2.

### 3.6. On the Prototype Loss for Trajectory Learning

Prototype learning can be seen as the combination of clustering and contrastive learning. Formally, for data points $X = \{x_1, ..., x_n\}$, the embedding function $f : \mathbb{R}^d \to \mathbb{R}^m$, and prototypes $\{p_1, ..., p_K\}$ are optimized by,

$$\min_{f, \{p_k\}} \sum_{i=1}^{n} \|f(x_i) - p_{y_i}\|^2 + \lambda \ell_{\text{contrast}}(f(x_i), p_{y_i}, \{p_k\}).$$

In trajectory imputation scenario, we propose two basic assumptions which ensure the representiveness of macro-level human movement patterns: (1) Human trajectory data is drawn from a mixture of distributions, each localized on a manifold region $\mathcal{M}$ with mean $\mu_k$. (2) The embedding

$f$ enables diverse prototypes that capture local tangents and reconstruct manifold structures via linear combinations (Roweis & Saul, 2000). With these assumptions, we have:

**Theorem 3.4** (The Optimality of Prototype Learning). *Any global optimum $(f^*, \{p_k^*\})$ satisfies:*

1. *Prototypes approximate conditional expectations: $p_k^* \approx \mathbb{E}\left[f^*(x)|x \in C_k\right]$.*

2. *Contrastive loss enforces prototype separation, forming diverse directional vectors: $\langle p_i^*, p_j^* \rangle \leq \epsilon, \text{ for } i \neq j$.*

*Proof.* Using Pollard's consistency theorem (Pollard, 1981), the empirical cluster centers converge to conditional expectations:

$$p_k^* \approx \mathbb{E}\left[f^*(x)|x \in C_k\right].$$

From InfoNCE-based contrastive loss (Saunshi et al., 2019), optimality conditions ensure prototype distinctiveness:

$$f^*(x)^\top p_y^* - f^*(x)^\top p_k^* \geq \delta, \quad \forall k \neq y.$$

where $\delta > 0$ .Since the clustering term already guarantees that $p_y^* \approx \mathbb{E}\left[f^*(x) \mid x \in C_y\right]$, averaging over cluster $C_y$ gives:

$$\langle p_y^*, p_y^* \rangle - \langle p_y^*, p_k^* \rangle \geq \delta.$$

shifting the terms leads to the observation:

$$\langle p_y^*, p_k^* \rangle \leq \|p_y^*\|^2 - \delta \leq \epsilon, k \neq y,$$

which indicates that contrastive loss forces prototypes into globally distinct directions, ensuring effective representation of manifold local structures. □

## 4. Experiments

### 4.1. Datasets

Our experiments utilize two well-established trajectory datasets: (1) WuXi: Extracted from mobile signal data (Song et al., 2017), covering WuXi, China, over six months (Oct 2013–Mar 2014). It records locations whenever users' phones are active. For efficiency, we use a 10-day subset, concatenating individual trajectories. (2) Foursquare: A public dataset (Yang et al., 2014) containing check-ins over 10 months (Apr 2012–Feb 2013) in New York and Tokyo. Each check-in includes a timestamp, GPS coordinates, and semantic tags. All datasets are anonymized, ensuring no privacy concerns. Tab. 6 in Appendix D provides details.

### 4.2. Evaluation Metric and Baseline

We evaluate trajectory imputation by comparing against (i) time-series interpolation methods and (ii) trajectory-specific approaches, with corresponding evaluation metrics.

*Table 1.* Comparison of model performance for different thresholds and different trajectory lengths on WuXi and FourSquare.

| | Method | WuXi | | | | | FourSquare | | | | |
|---|---|---|---|---|---|---|---|---|---|---|---|
| | | TC@2k | TC@4k | TC@6k | TC@8k | TC@10k | TC@2k | TC@4k | TC@6k | TC@8k | TC@10k |
| k=4 | VAR (Lütkepohl, 2013) | 0.5194 | 0.5632 | 0.6050 | 0.6441 | 0.6811 | 0.5000 | 0.5000 | 0.5000 | 0.5000 | 0.5000 |
| | SAITS (Du et al., 2023) | 0.5059 | 0.5224 | 0.5498 | 0.5861 | 0.6311 | 0.5000 | 0.5000 | 0.5000 | 0.5000 | 0.5000 |
| | TimesNet (Wu et al., 2023) | 0.5080 | 0.5290 | 0.5593 | 0.5955 | 0.6352 | 0.5015 | 0.5054 | 0.5133 | 0.5258 | 0.5431 |
| | Diff-TS (Yuan & Qiao, 2024) | 0.5123 | 0.5462 | 0.5951 | 0.6496 | 0.7060 | 0.5268 | 0.5714 | 0.6173 | 0.6571 | 0.6932 |
| | DiffTraj (Zhu et al., 2024a) | 0.6958 | 0.8198 | 0.8816 | 0.9169 | 0.9402 | 0.5945 | 0.6845 | 0.7574 | 0.8189 | 0.8666 |
| | Diff+Mask (Ours) | 0.6584 | 0.7731 | 0.8400 | 0.8884 | 0.9159 | 0.6541 | 0.7379 | 0.8010 | 0.8525 | 0.8928 |
| | ProDiff (Ours) | **0.7155** | **0.8414** | **0.9006** | **0.9326** | **0.9520** | **0.6644** | **0.7452** | **0.8087** | **0.8596** | **0.8971** |
| k=6 | VAR (Lütkepohl, 2013) | 0.3360 | 0.3437 | 0.3556 | 0.3692 | 0.3840 | 0.3333 | 0.3333 | 0.3334 | 0.3334 | 0.3335 |
| | SAITS (Du et al., 2023) | 0.3427 | 0.3762 | 0.4275 | 0.4880 | 0.5533 | 0.3333 | 0.3333 | 0.3333 | 0.3333 | 0.3333 |
| | TimesNet (Wu et al., 2023) | 0.3419 | 0.3654 | 0.4029 | 0.4500 | 0.5044 | 0.3386 | 0.3530 | 0.3756 | 0.4039 | 0.4341 |
| | Diff-TS (Yuan & Qiao, 2024) | 0.3515 | 0.4011 | 0.4726 | 0.5491 | 0.6211 | 0.3761 | 0.4283 | 0.4827 | 0.5383 | 0.5874 |
| | DiffTraj (Zhu et al., 2024a) | 0.5976 | 0.7476 | 0.8227 | 0.8688 | 0.9005 | 0.4277 | 0.5404 | 0.6428 | 0.7314 | 0.8025 |
| | Diff+Mask (Ours) | 0.5767 | 0.7324 | 0.8228 | 0.8802 | 0.9180 | 0.4859 | 0.5970 | 0.6902 | 0.7671 | 0.8265 |
| | ProDiff (Ours) | **0.5978** | **0.7686** | **0.8518** | **0.8992** | **0.9285** | **0.5005** | **0.6093** | **0.7013** | **0.7772** | **0.8345** |
| k=8 | VAR (Lütkepohl, 2013) | 0.2537 | 0.2627 | 0.2739 | 0.2861 | 0.2986 | 0.2500 | 0.2500 | 0.2500 | 0.2500 | 0.2500 |
| | SAITS (Du et al., 2023) | 0.2572 | 0.2764 | 0.3059 | 0.3485 | 0.3976 | 0.2500 | 0.2502 | 0.2505 | 0.2509 | 0.2513 |
| | TimesNet (Wu et al., 2023) | 0.2520 | 0.2574 | 0.2663 | 0.2785 | 0.2942 | 0.2516 | 0.2563 | 0.2634 | 0.2715 | 0.2808 |
| | Diff-TS (Yuan & Qiao, 2024) | 0.2689 | 0.3199 | 0.3907 | 0.4676 | 0.5453 | 0.3233 | 0.3932 | 0.4611 | 0.5358 | 0.5964 |
| | DiffTraj (Zhu et al., 2024a) | 0.5418 | 0.7009 | 0.7868 | 0.8414 | 0.8795 | 0.3316 | 0.4526 | 0.5671 | 0.6688 | 0.7520 |
| | Diff+Mask (Ours) | 0.4486 | 0.5946 | 0.6943 | 0.7631 | 0.8107 | 0.3957 | 0.5300 | 0.6431 | 0.7350 | 0.8045 |
| | ProDiff (Ours) | **0.5752** | **0.7501** | **0.8236** | **0.8663** | **0.8945** | **0.4000** | **0.5331** | **0.6474** | **0.7404** | **0.8090** |
| k=10 | VAR (Lütkepohl, 2013) | 0.2012 | 0.2047 | 0.2102 | 0.2177 | 0.2270 | 0.2000 | 0.2000 | 0.2000 | 0.2000 | 0.2000 |
| | SAITS (Du et al., 2023) | 0.2080 | 0.2316 | 0.2686 | 0.3158 | 0.3692 | 0.2000 | 0.2000 | 0.2000 | 0.2000 | 0.2000 |
| | TimesNet (Wu et al., 2023) | 0.2073 | 0.2275 | 0.2591 | 0.3035 | 0.3559 | 0.2003 | 0.2013 | 0.2034 | 0.2064 | 0.2110 |
| | Diff-TS (Yuan & Qiao, 2024) | 0.2173 | 0.2655 | 0.3367 | 0.4190 | 0.5000 | 0.2751 | 0.3484 | 0.4207 | 0.4990 | 0.5646 |
| | DiffTraj (Zhu et al., 2024a) | 0.4994 | 0.6687 | 0.7640 | 0.8259 | 0.8692 | 0.2762 | 0.4024 | 0.5300 | 0.6453 | 0.7386 |
| | Diff+Mask (Ours) | 0.3793 | 0.5104 | 0.6046 | 0.6773 | 0.7344 | 0.3412 | 0.4800 | 0.5999 | 0.7023 | 0.7868 |
| | ProDiff (Ours) | **0.4996** | **0.6994** | **0.8048** | **0.8667** | **0.9053** | **0.3522** | **0.4910** | **0.6105** | **0.7146** | **0.7920** |

For time-series interpolation, we benchmark against classical methods like Vector Autoregression (VAR) (Lütkepohl, 2005) and state-of-the-art spatio-temporal models, including SAITS (Du et al., 2023), TimesNet (Wu et al., 2023), Diffusion-TS (Yuan & Qiao, 2024), and DiffTraj (Zhu et al., 2024a). To measure imputation accuracy, we introduce trajectory coverage ($TC@\tau$), which quantifies the proportion of generated points $\hat{\mathbf{s}}_{i,j}$ within a threshold $\tau$ from the ground truth $\mathbf{s}_{i,j}$:

$$TC@\tau = \frac{1}{k} \sum_{j=1}^{k} \mathbb{I}\left(d(\hat{\mathbf{s}}_{i,j}, \mathbf{s}_{i,j}) < \tau\right), \quad (15)$$

where $\hat{\mathbf{s}}_{i,j}$ is the generated points, and $\mathbb{I}(\cdot)$ is an indicator function that equals when $d(\hat{\mathbf{s}}_{i,j}, \mathbf{s}_{i,j})$ is less than the threshold $\tau$ and 0 otherwise.

For trajectory-specific baselines, we compare against RN-TrajRec (Chen et al., 2023), TS-TrajGen (Jiang et al., 2023b), MM-STGED (Wei et al., 2024), AttnMove (Xia et al., 2021), and DiffTraj (Zhu et al., 2024a). To ensure fairness, we remove modules reliant on unavailable auxiliary information. Performance is assessed using standard trajectory generation metrics, including Density, Distance, Segment Distance, Radius, MAE, and RMSE. Further details on evaluation protocols and baselines are provided in Appendix D.

### 4.3. Implementation Details

Our experiments balance the effectiveness of each module in the joint training; we set $\lambda_1, \lambda_2, \lambda_3$ to 1. Gradient updates were facilitated using the Adam optimizer, initialized with a learning rate of $2e^{-4}$. We summarize the hyperparameter settings for the diffusion model and PCE in Tab. 2.

*Table 2.* General setting of ProDiff model

| Diffusion | | PCE | |
|---|---|---|---|
| Parameter | Setting value | Parameter | Setting value |
| Diffusion Steps | 500 | Prototypes | 20 |
| Embedding Dim | 128 | Embedding Dim | 512 |
| $\beta$ (linear schedule) | $0.0001 \sim 0.05$ | Heads | 8 |
| ResNet Blocks | 2 | Encoder Blocks | 4 |
| Sampling Blocks | 4 | Forward Dim | 256 |
| Input Length | $3 \sim 10$ | Dropout ratio | 0.1 |

### 4.4. Main Results

The trajectory coverage across different baselines and window sizes is presented in Tab. 1, where "TC@2k" represents the percentage of generated values within 2km of the true location, with TC@4k–10k extending up to 10km. The highest and second-highest values are marked in red and blue, respectively. We evaluate performance separately for time-series interpolation methods and trajectory-specific approaches.

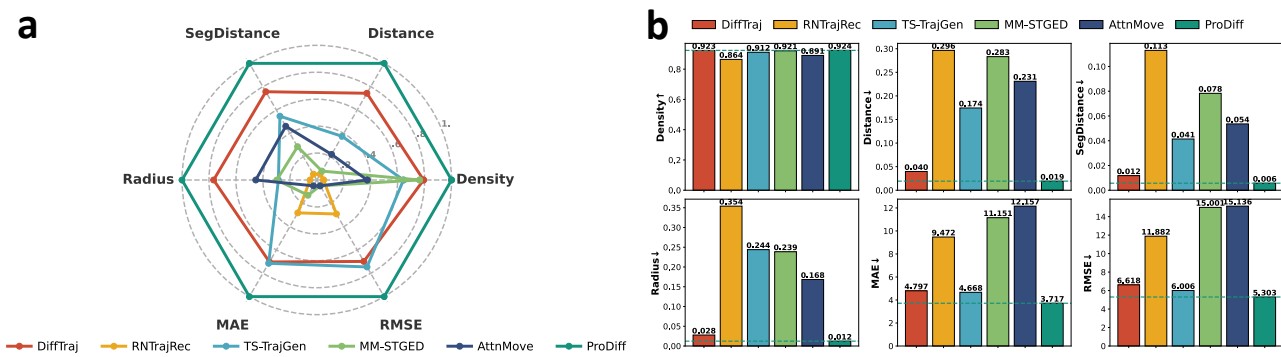

*Figure 4.* **a**. Radar charts illustrate the normalized performance of different models across six distinct metrics. **b**. Histogram comparing the performance of each model across different metrics, with dashed lines indicating the best-performing model's values for each metric.

**(1) Comparison with Time-Series Interpolation Methods.** ProDiff consistently outperforms sequence imputation models across datasets. On the WuXi dataset with $k = 4$, ProDiff achieves 71.55% at TC@2k, exceeding all baselines ($<$70%). As the threshold increases (TC@4k–10k), ProDiff maintains high accuracy (84.14%–95.20%), with its advantage over the second-best method expanding from 1.18% at TC@2k to 3.61% at TC@10k. Furthermore, ProDiff demonstrates robustness across datasets and segment sizes, where other models, such as DiffTraj, suffer sharp declines at larger thresholds (TC@6k–10k). On the FourSquare dataset, DiffTraj's TC@2k score for $k = 8$ drops by 21.02%, while ProDiff only decreases by 11.59%. Additionally, while Diff-Traj loses its second-place ranking to Diff+Mask, ProDiff retains its lead, indicating its ability to learn stable movement patterns via the prototype condition extractor.

**(2) Comparison with Trajectory-Specific Methods.** Fig. 4 further validates ProDiff's superiority among trajectory models. Panel (a) presents normalized scores across all metrics, while panel (b) details model-specific performance. ProDiff consistently sets the benchmark across six additional metrics. While density scores are similar among models, ProDiff exhibits a substantial lead in spatial distribution metrics (e.g., Distance, Segment Distance, Radius, MAE, RMSE), highlighting its effectiveness in diverse conditions.

### 4.5. Ablation Study

We conducted three ablation experiments on the WuXi dataset to validate the contributions of our key components. We also investigate the accerlaration of the proposed ProDiff, and the results are provided in Appendix D.

**Effect of Prototype Condition Extractor.** To assess the impact of individual modules, we removed the prototype condition extractor (PCE), cross-entropy loss ($\mathcal{L}_{C1}$), and contrastive loss ($\mathcal{L}_{C2}$) while keeping the joint loss intact. As shown in Tab. 3, PCE consistently improves performance,

with $\mathcal{L}_{C1}$ and $\mathcal{L}_{C2}$ further enhancing its effectiveness in capturing movement patterns. Notably, the performance gain of PCE is more pronounced at longer distances when $k = 8$, suggesting that its effectiveness increases with extended trajectory segments.

*Table 3.* Performance comparison of removing different modules.

| | Method | TC@2k | TC@4k | TC@6k | TC@8k | TC@10k |
|---|---|---|---|---|---|---|
| $k=6$ | ProDiff | **0.5978** | **0.7686** | **0.8518** | **0.8992** | **0.9285** |
| | w.o. Pro | 0.5767 | 0.7324 | 0.8228 | 0.8802 | 0.9180 |
| | w.o. $\mathcal{L}_{C1}$ | 0.5939 | 0.7556 | 0.8371 | 0.8867 | 0.9195 |
| | w.o. $\mathcal{L}_{C2}$ | 0.5952 | 0.7560 | 0.8374 | 0.8869 | 0.9199 |
| $k=8$ | ProDiff | **0.5752** | **0.7501** | **0.8236** | **0.8663** | **0.8945** |
| | w.o. Pro | 0.4486 | 0.5946 | 0.6943 | 0.7631 | 0.8107 |
| | w.o. $\mathcal{L}_{C1}$ | 0.5395 | 0.7205 | 0.7966 | 0.8399 | 0.8691 |
| | w.o. $\mathcal{L}_{C2}$ | 0.4888 | 0.6638 | 0.7473 | 0.7984 | 0.8340 |

*Table 4.* Performance comparison for cVAE and cGAN.

| Method | TC@2k | TC@4k | TC@6k | TC@8k | TC@10k |
|---|---|---|---|---|---|
| cVAE+MASK | 0.2616 | 0.2936 | 0.3385 | 0.3926 | 0.4513 |
| cVAE+pro | 0.3416 | 0.3685 | 0.4082 | 0.4540 | 0.5009 |
| cGAN+MASK | 0.2760 | 0.3240 | 0.3742 | 0.4285 | 0.4896 |
| cGAN+pro | 0.3074 | 0.3997 | 0.4746 | 0.5361 | 0.5883 |

**Generalization Across Generative Models.** To evaluate PCE's generalizability, we integrated it with cVAE and cGAN, applying both MASK and PCE to these models (Tab. 4). At the 10k threshold, adding only MASK yields 45.13% for cVAE and 48.96% for cGAN, whereas incorporating PCE improves accuracy to 50.09% and 58.83%, respectively. This highlights PCE's ability to enhance movement pattern learning across different generative frameworks.

### 4.6. Utility of Generated Data

To evaluate the real-world applicability of ProDiff generated data, we tested its performance on traffic flow analysis in

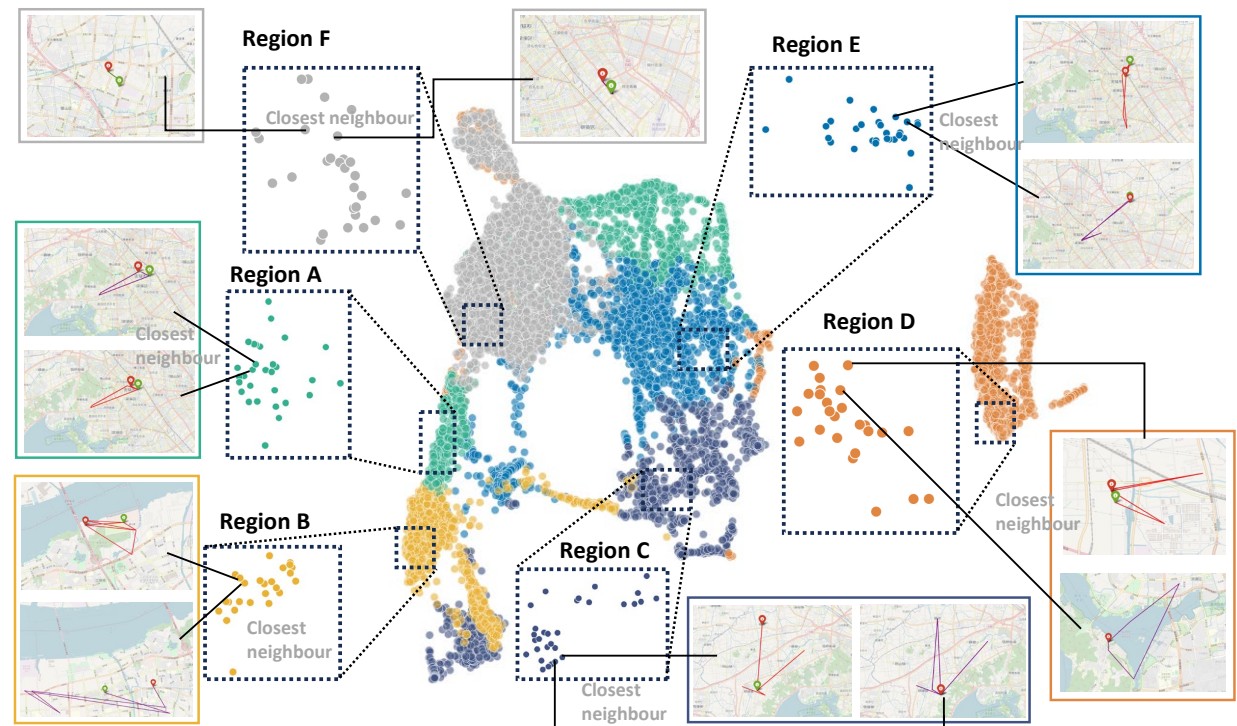

*Figure 5.* Trajectory data representation after dimensionality reduction by PaCMAP, randomly selected samples and neighboring samples plot trajectories to interpret human trajectory patterns captured by prototype learning.

WuXi, using $k = 6$ trajectory imputations over 7000 individuals across 10 days. The city was divided into 1km × 1km grids (longitude gap $\approx 0.009°$), where each grid's value increments as individuals' trajectories pass through. Fig. 6(a) (top) compares real and ProDiff-generated traffic maps, revealing highly similar spatial patterns. To further analyze peak and trend consistency, we extracted and projected traffic edges (Fig. 6(a), bottom), showing near-identical fluctuations between real and generated data. Additionally, correlation coefficients and spatial distributions between real and generated data (Fig. 6(b), 6(c)) further confirm the reliability of ProDiff's imputation. These results demonstrate that ProDiff can generate realistic and usable trajectory data, making it applicable to downstream mobility analysis tasks.

*Table 5.* Impact of different numbers of prototypes (N) and trajectory length (k).

|   | N | TC@2k | TC@4k | TC@6k | TC@8k | TC@10k |
|---|---|-------|-------|-------|-------|--------|
| k=6 | 15 | 0.5881 | 0.7583 | 0.8433 | 0.8928 | 0.9237 |
|   | 20 | 0.5978 | 0.7686 | 0.8518 | 0.8992 | 0.9285 |
|   | 25 | 0.5868 | 0.7570 | 0.8441 | 0.8951 | 0.9265 |
| k=8 | 15 | 0.5755 | 0.7552 | 0.8217 | 0.8763 | 0.8951 |
|   | 20 | 0.5752 | 0.7501 | 0.8236 | 0.8663 | 0.8945 |
|   | 25 | 0.5785 | 0.7553 | 0.8237 | 0.8634 | 0.9017 |

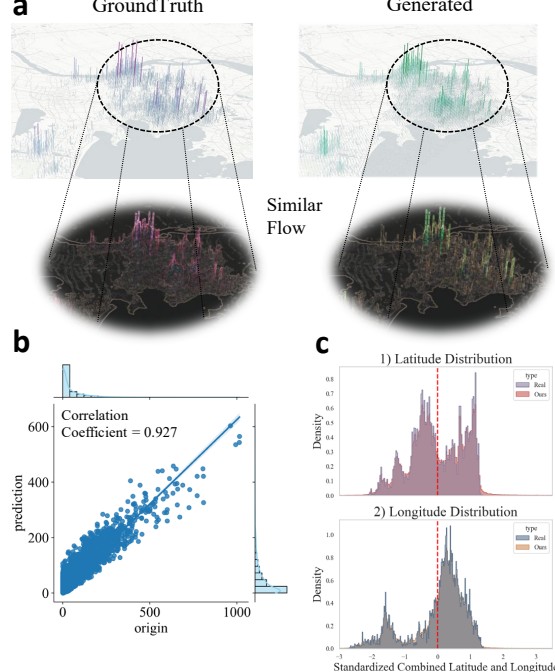

*Figure 6.* a. Comparison of traffic patterns between groundtruth and generated data. b. The correlation coefficient between groundtruth and generated data. c. Comparison of spatial distributions after normalization of both real and generated data.

### 4.7. Hyperparameter Sensitivity.

We analyze the effect of prototype count ($N$), trajectory length ($k$), and diffusion steps ($d$) (Tab. 5, Appendix Tab. 7). Increasing $N$ from 15 to 20 improves TC@10k to 0.9285 for $k = 6$, while $k = 8$ benefits from $N = 25$, suggesting behavioral variations across window sizes. Diffusion steps significantly affect performance, with 300 steps yielding optimal TC@10k (0.9300). Beyond this, performance plateaus, while computational cost increases, making 300 steps a practical balance between accuracy and efficiency.

### 4.8. Interpretability Analysis

Understanding whether prototype learning captures interpretable movement patterns in low-dimensional space is essential for evaluating the effectiveness of the joint prototype learning-diffusion framework. Fig. 5 visualizes this process, where trajectory data is fed into the trained prototype condition extractor, clustered using K-means (top 6 classes), and reduced in dimensionality via PaCMAP. To interpret the latent space, we zoom into each class, plotting selected samples and their nearest neighbors. The learned movement patterns exhibit clear semantic coherence. Region A captures trajectories with start and end points in close proximity, reflecting movement within similar locations. Region B extends this pattern, with slightly farther start and end points, aligning with the proximity of the yellow and green clusters. Region C represents trajectories constrained within similar locations. Region D deviates from previous patterns, showing long-distance migration with a return to the starting point. Region E follows a linear migration and return pattern. Region F is similar to Region A, but its neighboring trajectories occur in different locations. These findings demonstrate the model's ability to embed human trajectory, capture structured movement behaviors, distinguish variations, and optimize representations during training, improving trajectory imputation performance when integrated with the diffusion framework.

## 5. Conclusion

This paper addresses the trajectory imputation problem, focusing on generating realistic trajectories with minimal information. Unlike conventional methods that rely on sparse trajectory pattern, we propose ProDiff, a prototype-guided diffusion model that captures macro-level mobility patterns while maintaining high fidelity in trajectory generation. Our experiments demonstrate that ProDiff outperforms state-of-the-art approaches on two datasets, improving trajectory imputation accuracy. Ablation studies confirm that prototype learning significantly enhances trajectory representation, while diffusion modeling effectively reconstructs realistic movements. Beyond imputation, ProDiff may be generalized to broader trajectory-related tasks, offering a scalable solution for urban mobility analysis and behavioral modeling. Moving forward, we aim to extend ProDiff to adaptive and personalized trajectory generation, integrating reinforcement learning and uncertainty-aware models to enhance reliability under dynamic and noisy conditions.

## Acknowledgments

Prof. Xin Lu's work was supported by the National Natural Science Foundation of China (72025405, 72421002, 92467302, 72474223, 72301285), the Science and Technology Innovation Program of Hunan Province (2023JJ40685, 2024RC3133), and the Major Program of Xiangjiang Laboratory (24XJJCYJ01001). Prof. Jingyuan Wang's work was partially supported by the National Natural Science Foundation of China (No. 72222022, 72171013). Suoyi Tan was supported by the National Nature Science Foundation of China (No. 72474223, 72001211 ), the science and technology innovation Program of Hunan Province (No. 2024RC3133), and the National University of Defense Technology Cornerstone Project (No. JS24-04).

## Impact Statement

Trajectory Imputation is essential for dealing with incomplete trajectory data, a common issue stemming from device limitations and varied collection scenarios. Our work presents ProDiff, a prototype-guided diffusion model, to effectively impute trajectories using only minimal information. This approach allows for robust spatiotemporal reconstruction of human movement, even from highly sparse data. While there will be important impacts resulting from improved trajectory imputation in general, here we focus on the impact of using our ProDiff framework for minimal information imputation. There are many benefits to using our method, such as significantly improving imputation accuracy and effectively embedding and capturing macro-level human movement patterns. This paper presents work whose goal is to advance the field of Trajectory Data Mining and Spatio-temporal Data Analysis. There are many potential societal consequences of our work, none which we feel must be specifically highlighted here.

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

# A. Detailed Related Work

**Spatial-Temporal Sequence Imputation.** Imputation methods for spatial-temporal sequences can be broadly divided into two categories: traditional methods and deep learning methods. Traditional methods include linear interpolation (Blu et al., 2004) and mean value filling, which are fast but overly simplistic and struggle to estimate the overall data distribution (Huang et al., 2023). More advanced probabilistic methods, such as probabilistic PCA (Qu et al., 2009) and expectation maximization (Shi et al., 2013), aim to capture the data distribution more accurately. Autoregressive methods like Vector Autoregressive (VAR) (Lütkepohl, 2013) and matrix/tensor-based methods (e.g., Tucker decomposition (Tan et al., 2013)) have also been used to address missing data in spatial-temporal contexts.

Deep learning-based imputation methods can be further divided into non-generative and generative approaches. Non-generative methods primarily rely on RNNs and attention mechanisms. For example, GRU-D (Che et al., 2018) proposes a variant of the gated recurrent unit (GRU) to handle missing data in time series, while TimesNet (Wu et al., 2023) leverages 2D convolutional neural networks to model temporal dependencies. Attention-based methods, such as CDSA (Ma et al., 2019) and SAITS (Du et al., 2023), focus on capturing both short-term and long-term dependencies across multiple dimensions (time, location, measurement). Generative methods include variational autoencoders (VAEs) (Doersch, 2016), generative adversarial networks (GANs) (Goodfellow et al., 2020), and diffusion probabilistic models, which have become increasingly popular. For instance, Diffusion-TS (Yuan & Qiao, 2024) combines diffusion models with time series decomposition to address missing data.

**Trajectory Data Mining.** Trajectory data mining based on deep learning methods can be categorized into several tasks, including trajectory forecasting, travel time estimation, and anomaly detection (Chen et al., 2024). Trajectory forecasting involves predicting future locations(Wang et al., 2021; Wu et al., 2019) or traffic conditions(Wu et al., 2020; Liu et al., 2024; Ji et al., 2023; Jiang et al., 2023a; Ji et al., 2022a). Common approaches include CNN-based models (Bao et al., 2021) and RNN-based models (Yang et al., 2017; Yao et al., 2017), with recent advances exploring diffusion techniques like BCDiff (Li et al., 2023) that bidirectionally refine historical and future trajectories through coupled diffusion models with adaptive gating mechanisms. Travel Time Estimation (TTE) or Estimated Time of Arrival (ETA) involves analyzing trajectory sequences to predict travel time. For example, eRCNN (Wang et al., 2016) uses raw GPS data with a recurrent convolutional neural network to estimate travel time and speed. Road-based TTE approaches, such as WDR (Wang et al., 2018), model the correlation between trips and roads using a regression framework.

Trajectory anomaly detection aims to identify abnormal movement patterns. Offline detection methods, like ATD-RNN (Song et al., 2018), use RNNs with fully connected layers for anomaly detection. Online detection methods leverage reinforcement learning to model the transition probability between road segments, treating anomaly detection as a sequential decision problem (Chen et al., 2024). GM-VSAE (Liu et al., 2020) adapts an RNN-based VAE model to learn the probability distribution in the latent space.

Recent advancements address data incompleteness challenges through unified frameworks, GC-VRNN (Xu et al., 2023) pioneers joint trajectory imputation and prediction using multi-space graph neural networks to capture spatio-temporal missing patterns and temporal decay modules for information recovery. Mobility generation tasks have also gained attention, with models like DiffTraj (Zhu et al., 2024a) utilizing diffusion models to generate synthetic trajectories at the population level. Our proposed task combines trajectory forecasting and mobility generation, where generative trajectory interpolation aligns with mobility generation, and generalized trajectory interpolation is considered a higher-order forecasting task.

**Mobility Data Synthesizing.** The generation of synthetic mobility data has been extensively studied to address privacy concerns, data scarcity, and high collection costs (Jia et al., 2020; Zhang et al., 2023; Zheng et al., 2014). Early non-generative approaches primarily relied on statistical models (Simini et al., 2021; Wang et al., 2019), perturbation techniques (Zandbergen, 2014), or simulations (Simini et al., 2021). While these methods offer insights into movement dynamics, they often fail to capture complex spatial-temporal relationships in real-world scenarios (Pappalardo et al., 2023).

With advancements in deep learning, generative approaches have gained prominence. Variational Autoencoders (VAEs) like TrajVAE (Chen et al., 2021) leverage temporal dependencies to produce realistic trajectories, while GAN-based frameworks such as TS-TrajGen (Jiang et al., 2023b) use coarse-to-fine modeling to generate synthetic trajectories from spatial grid transformations. However, these models often face limitations in achieving high-resolution fidelity, particularly when translating grid-based representations into fine-grained data.

Graph-based approaches have been widely investigated due to their ability to capture spatial-temporal relationships effectively. For example, RNTrajRec (Chen et al., 2023) employs a graph-based framework that integrates graph representations of

trajectory points and spatial-temporal transformers to model dependencies along the trajectory, significantly enhancing trajectory recovery accuracy. Similarly, MM-STGED (Wei et al., 2024) utilizes a graph-based encoder-decoder structure to represent trajectories as spatial-temporal graphs, capturing both micro-level semantics of GPS points and macro-level semantics of shared travel patterns.

Attention-based architectures, such as AttnMove (Xia et al., 2021), leverage attention mechanisms to model spatial-temporal correlations explicitly, facilitating the reconstruction of missing trajectory data and improving performance in downstream applications. Recent innovations in trajectory generation include the use of denoising diffusion probabilistic models (DDPMs) (Ho et al., 2020), which iteratively refine noisy inputs to produce high-fidelity synthetic data. For instance, DiffTraj (Zhu et al., 2024a) captures spatial-temporal dependencies without relying on intermediate transformations, offering significant advantages in privacy preservation and data utility. Additionally, ControlTraj (Zhu et al., 2024b) extends the diffusion framework by integrating conditional signals for controllable generation, improving its applicability across varied scenarios.

## B. Detailed Denoising Network

### B.1. Denoising Diffusion Probabilistic Model

The diffusion probabilistic model has gained increasing attention in recent years for its success in various data generation tasks. The model consists of a *forward process* that gradually perturbs the data distribution with noise, and a *reverse (denoising) process* that learns to reconstrust the original data distribution.

**Forward Process.** Given a set of data samples $\boldsymbol{x}_0 \sim q(\boldsymbol{x}_0)$, the forward process adds $T$ time-steps of Gaussian noise $\mathcal{N}(\cdot)$ to it, where $T$ is an adjustable parameter. Formally, the forward process can be defined as a Markov chain from data $\boldsymbol{x}_0$ to the latent variable $\boldsymbol{x}_T$:

$$q(\boldsymbol{x}_{1:T} \mid \boldsymbol{x}_0) = \prod_{t=1}^{T} q(\boldsymbol{x}_t \mid \boldsymbol{x}_{t-1}) \tag{16}$$

$$q(\boldsymbol{x}_t \mid \boldsymbol{x}_{t-1}) = \mathcal{N}(\boldsymbol{x}_t; \sqrt{1 - \beta_t}\boldsymbol{x}_{t-1}, \beta_t \boldsymbol{I}), \tag{17}$$

in which $\{\beta_t \in (0,1)\}_{t=1}^{T}(\beta_1 < \beta_2 < ... < \beta_T)$ is the corresponding variance schedule. Since it is impractical to back-propagate the gradient by sampling from a Gaussian distribution, we adopt a parameterization trick to keep the gradient derivable and the $\boldsymbol{x}_t$ can be expressed as $\boldsymbol{x}_t = \sqrt{\bar{\alpha}_t}\boldsymbol{x}_0 + \sqrt{1 - \bar{\alpha}_t}\boldsymbol{\epsilon}$, where $\boldsymbol{\epsilon} \sim \mathcal{N}(0, \boldsymbol{I})$ and $\bar{\alpha}_t = \prod_{i=1}^{t}(1 - \beta_i)$.

**Reverse Process.** The reverse diffusion process, also known as the denoising processing, aims to reconstruct the original data distribution from the noisy data $\boldsymbol{x}_T \sim \mathcal{N}(0, \boldsymbol{I})$. Accordingly, this process can be formulated by the following Markov chain:

$$p_\theta(\boldsymbol{x}_{0:T}) = p(\boldsymbol{x}_T) \prod_{t=1}^{T} p_\theta(\boldsymbol{x}_{t-1} \mid \boldsymbol{x}_t) \tag{18}$$

$$p_\theta(\boldsymbol{x}_{t-1} \mid \boldsymbol{x}_t) = \mathcal{N}\left(\boldsymbol{x}_{t-1}; \boldsymbol{\mu}_\theta(\boldsymbol{x}_t, t), \boldsymbol{\sigma}_\theta(\boldsymbol{x}_t, t)^2 \boldsymbol{I}\right), \tag{19}$$

where $\boldsymbol{\mu}_\theta(\boldsymbol{x}_t, t)$ and $\boldsymbol{\sigma}_\theta(\boldsymbol{x}_t, t)$ are the mean and variance parameterized by $\theta$, respectively. Based on the literature, for any $\tilde{\beta}_t = \frac{1 - \bar{\alpha}_{t-1}}{1 - \bar{\alpha}_t}\beta_t(t > 1)$ and $\tilde{\beta}_1 = \beta_1$, the parameterizations of $\boldsymbol{\mu}_\theta$ and $\boldsymbol{\sigma}_\theta$ are defined by:

$$\boldsymbol{\mu}_\theta(\boldsymbol{x}_t, t) = \frac{1}{\sqrt{\alpha_t}}\left(\boldsymbol{x}_t - \frac{\beta_t}{\sqrt{1 - \bar{\alpha}_t}}\boldsymbol{\epsilon}(\boldsymbol{x}_t, t)\right), \boldsymbol{\sigma}_\theta(\boldsymbol{x}_t, t) = \tilde{\beta}_t^{\frac{1}{2}}. \tag{20}$$

# C. Method

---

**Algorithm 1** Training of ProDiff

---

    **for** $i = 1, 2, \ldots,$ **do**
        Get base condition $\boldsymbol{B}^c$
        Get prototype condition $\boldsymbol{P}^c$ by PCE network
        Get $WD(\boldsymbol{B}^c), WD(\boldsymbol{P}^c)$ by Wide & Deep network
        Get conditional guidance $\mathcal{J}^c = WD(\boldsymbol{B}^c) + WD(\boldsymbol{P}^c)$
        $f_\gamma(\mathbf{Z}_0) = \mathcal{J}^c$
        Sample $\mathbf{Z} \sim p$ where $p$ represents the distribution of original data
        Sample $t \sim \mathcal{U}[0, T], \epsilon \sim \mathcal{N}(0, \mathbf{I}_{l \times d})$
        $\mathbf{Z}_t = \sqrt{\bar{\alpha}_t}\mathbf{Z}_0 + \sqrt{1 - \bar{\alpha}_t}\epsilon$
        Updating the gradient $\nabla_{\theta/\gamma}\mathcal{L}_J$ which means optimizing $\mathbb{E}_{t \sim \mathcal{U}[0,T]}\mathbb{E}_{\mathbf{Z}_0 \sim p, \epsilon \sim \mathcal{N}} \left[ \nabla_{\theta/\gamma} \| \epsilon - \epsilon_\theta(\mathbf{Z}_t, t, f_\gamma(\mathbf{Z}_0)) \|^2 \right]$
    **end for**

---

---

**Algorithm 2** Sampling of ProDiff

---

1: Get data and Sample $\tilde{\mathbf{Z}}_T \sim \mathcal{N}(0, \mathbf{I})$
2: Get base condition $\boldsymbol{B}^c$
3: Get prototype condition $\boldsymbol{P}^c$ by PCE network
4: Get $WD(\boldsymbol{B}^c), WD(\boldsymbol{P}^c)$ by Wide & Deep network
5: Get conditional guidance $\mathcal{J}^c = WD(\boldsymbol{B}^c) + WD(\boldsymbol{P}^c)$
6: **for** $t = T, T - S, \ldots, 1$ **do**
7:    Compute $\mu_\theta\left(\tilde{\mathbf{Z}}_t, t, \mathcal{J}^c\right)$ according to Eq.(20)
8:    Compute $p_\theta\left(\tilde{\mathbf{Z}}_{t-1} \mid \tilde{\mathbf{Z}}_t, \mathcal{J}^c\right)$ according to Eq.(19)
9: **end for**
10: **Return:** $\tilde{\mathbf{Z}}_0$

---

# D. Experiment

## D.1. Dataset

We evaluate the performance of ProDiff and all baselines methods on two datasets, **WuXi** and **FourSquare**.

The mobile phone dataset WuXi used in this study were collected between October 24, 2013, and March 24, 2014, in Wuxi, China, encompassing approximately six million users evenly distributed across the area. Every hour, these users generate around 40 million raw records, each containing essential location information, including cell-id and area-id, which correspond to specific cell towers. Each record in the dataset includes four key components: user ID, cell tower ID, timestamp, and a tag. The timestamp indicates the exact moment the record was created, while the tag specifies the type of activity associated with the record. For the purpose of this study, we focused on data from ten consecutive days, concatenating individual trajectories during this period. This subset includes 33000 active users and 671,124 location updates, of which 30,000 users are used for training and 3,000 users for testing.

The FourSquare dataset contains Foursquare check-ins over ten months (from April 12, 2012, to February 16, 2013), filtered for noise and invalid check-ins. It includes active users in two major cities, New York and Tokyo, with each check-in associated with a timestamp, GPS coordinates, and semantic meaning. We did not use the taxi-related dataset because human trajectories have a higher degree of freedom compared to car trajectories. Due to the volume of data, only Tokyo data was used on the FourSquare dataset. Tab. 6 summarizes the statistical information of these two datasets, which includes 2,293 active users with 573,703 location updates.

## D.2. Preprocess

In trajectory data analysis, careful preparation of raw data is fundamental to ensure the reliability and precision of computational models. Our preprocessing approach transforms raw GPS coordinates into a format optimized for training

Table 6. Statistics of two human mobility datasets.

| Dataset | WuXi | FourSquare |
|---|---|---|
| Time Span (day) | 111 | 310 |
| Used Time Span (day) | 10 | 310 |
| Train Active Users | 30000 | 1834 |
| Test Active Users | 3000 | 459 |
| Location Updates | 671,124 | 573,703 |
| Average Distance (meter) | 3336.33 | 4301.51 |
| Average Time (hour) | 7.8 | 37.15 |

deep learning systems, focusing on two key steps: segmentation and normalization.

**Segmentation** involves dividing continuous trajectory data into fixed-length segments using a sliding window method. This technique incrementally generates samples from a single trajectory. Trajectories matching the target length are directly included as individual samples, while longer paths are systematically partitioned into uniform segments. This creates discrete, standardized inputs for model training.

**Normalization** adapts the data for diffusion-based models, which rely on introducing and removing Gaussian noise during training. To align with the noise distribution, spatial coordinates are scaled to a dimensionless, standardized range (e.g., $[0, 1]$). This eliminates scale variations between features, allowing the model to focus on spatial patterns rather than magnitude differences. Crucially, the process is fully reversible—after model inference, outputs can be rescaled to their original geographic coordinates, preserving real-world interpretability.

### D.3. Evaluation Metric and Baseline

The trajectory imputation task aims to fill in missing points as accurately as possible under the given point conditions. To assess performance, we propose a novel trajectory coverage metric, measuring the percentage of generated locations within a specified distance from the groundtruth. Given a threshold $\tau$, we count the number of generated points within $\tau$ distance from the groundtruth and divide by the total length of the trajectory to limit it to the $[0, 1]$ interval. For any trajectory $\mathbf{S}_i = [\mathbf{s}_{i,1}, \mathbf{s}_{i,2}, ..., \mathbf{s}_{i,k}]$ of length $k$, we construct a masked trajectory $\mathbf{S}_i^m = [\mathbf{s}_{i,1}, \mathbf{s}_{i,k}]$. Given this condition and threshold $\tau$, ProDiff generates the missing points, and we calculate the trajectory coverage $TC@\tau$ as the following equation:

$$TC@\tau = \frac{1}{k} \sum_{j=1}^{k} \mathbb{I}(d(\hat{\mathbf{s}_{i,j}}, \mathbf{s}_{i,j}) < \tau), \tag{21}$$

where $\hat{\mathbf{s}_{i,j}}$ is the generated points from ProDiff, and $\mathbb{I}(\cdot)$ is an indicator function that equals 1 when $d(\hat{\mathbf{s}_{i,j}}, \mathbf{s}_{i,j})$ is less than the threshold $\tau$ and 0 otherwise. We use the haversine function as the distance metric, which calculates the great-circle distance between two points on the Earth's surface, accurately reflecting the true distance by converting latitude and longitude into radians.

We select some traditional methods and the current state-of-the-art spatio-temporal sequence methods based on the Diffusion model, as well as trajectory-related methods to realize the trajectory imputation task. The compared baselines are set as follows:

**VAR**: Vector Autoregression (VAR) is a traditional model used to capture the linear interdependencies among multiple time series data. By considering each variable's own lagged values and the lagged values of other variables in the system, VAR models can effectively analyze the dynamic relationships and forecast future movements of the variables (Lütkepohl, 2005).

**SAITS**: SAITS is an advanced model(Du et al., 2023) designed to handling missing data in time series analysis. By leveraging self-attention mechanisms, SAITS aims to capture both short-term and long-term dependencies within time series data, This model stands out due to its ability to focus on the most relevant parts of the input data.

**TimesNet**: TimesNet is a progressive model (Wu et al., 2023) which treats time series data as 2D tensors, allowing it to leverage powerful 2D convolutional neural networks to model temporal dependencies. This approach is suitable for a wide range of applications, including missing value handling, forecasting, and anomaly detection.

**Diffusion-TS**: A current SOTA method (Yuan & Qiao, 2024) for time series generation task based on diffusion model and it also applies to missing value processing tasks. Diffusion-TS decomposes time series into interpretable variables, combining seasonal trend decomposition techniques and denoising diffusion models.

**DiffTraj**: Generating GPS Trajectory with Diffusion Probabilistic Model is a SOTA model(Zhu et al., 2024a) designed for generating realistic GPS trajectories. DiffTraj progressively refines random noise into coherent and plausible GPS trajectory data through a series of probabilistic steps. It is worth noting that while DiffTraj can also be used for the trajectory imputation task, it generates trajectories at the population level, which is fundamentally different from what we have done at the individual level.

We further evaluated our model against external baselines and metrics. Specifically, we incorporated RNTrajRec(Chen et al., 2023), TS-TrajGen(Jiang et al., 2023b), MM-STGED(Wei et al., 2024), and AttnMove(Xia et al., 2021) as baseline models:

**RNTrajRec**: RNTrajRec(Chen et al., 2023) employs a graph-based framework that integrates graph representations of trajectory points and spatial-temporal transformers to model dependencies along the trajectory, significantly enhancing trajectory recovery accuracy.

**TS-TrajGen**: TS-TrajGen(Jiang et al., 2023b) proposes a hierarchical generation framework that employs coarse-to-fine modeling to synthesize realistic trajectories. It first learns spatial grid-based latent representations to capture macroscopic movement patterns, then refines trajectories through adaptive spatial transformations and temporal interpolation. This approach effectively addresses the sparsity of raw GPS data while preserving topological consistency with road networks.

**MM-STGED**: MM-STGED(Wei et al., 2024) utilizes a graph-based encoder-decoder structure to represent trajectories as spatial-temporal graphs, capturing micro-level semantics of GPS points and macro-level semantics of shared travel patterns.

**AttnMove**: AttnMove(Xia et al., 2021), leverage attention mechanisms to model spatial-temporal correlations explicitly, facilitating the reconstruction of missing trajectory data and improving performance in downstream applications.

To ensure fair comparisons, specific modules in these baselines were removed to avoid reliance on unavailable additional information from our datasets. Additionally, we introduced extra metrics to assess the spatial distribution of generated trajectories:

- Density: Measures the cosine similarity of grid density between real and generated trajectories (higher is better).

- Distance: Evaluates the difference in travel distance between real and generated data, calculated as the sum of distances between consecutive points (lower is better).

- Segment Distance: Assesses the difference in segment distance between real and generated data, defined as the distance between consecutive points (lower is better).

- Radius: Evaluates the root mean square distance of all activity locations from the central location, indicating the spatial range (lower is better).

- MAE: Mean absolute error, measuring the average magnitude of errors between real and generated trajectories (lower is better).

- RMSE: Root mean square error, evaluating the square root of the average squared differences between predicted and actual values (lower is better).

### D.4. Exploratory Study

To assess ProDiff's performance under varying information conditions, we conducted two additional sets of experiments for the trajectory imputation task. Specifically, given a trajectory length of 10, in addition to fixing the begin and end points, supplementary information points were added to guide the model to accomplish better generation, as shown in the first four rows of Tab. 8. Meanwhile, motivated by the conclusion in the literature (De Montjoye et al., 2013) (that 95% of the personnel's trajectories can be determined by arbitrarily giving 4 points), we modify the information of the fixed points to randomly selecting x points during both training and testing. This scenario, represented in the last four rows of Tab. 8, is more challenging since the selected points vary and thus introduce more complexity. From the results of whole table, some key findings can be obtained: (i) Impact of Increased Information: The results show a significant improvement in trajectory imputation when the number of given points increases from 2 to 4. Beyond four points, the enhancement in performance

becomes marginal. (ii) Fixed vs. Random Points: When fixing four points, our method's performance aligns closely with the findings of (De Montjoye et al., 2013). However, when points are selected randomly, the model's performance diverges more noticeably from the literature's conclusion. This discrepancy likely arises because random points disrupt the consistency of trajectory sampling, increasing the difficulty of model learning. The results highlight the potential of ProDiff in accurately imputing missing trajectory points with a sufficient number of fixed reference points. However, the challenge remains when dealing with randomly selected points, indicating an area for future improvement.

*Table 7.* Impact of different diffusion steps (d).

| d | TC@2k | TC@4k | TC@6k | TC@8k | TC@10k |
|---|-------|-------|-------|-------|--------|
| 100 | 0.5750 | 0.7445 | 0.8329 | 0.8853 | 0.9187 |
| 300 | 0.6015 | 0.7697 | 0.8524 | 0.9005 | 0.9300 |
| 500 | 0.5978 | 0.7686 | 0.8515 | 0.8992 | 0.9285 |
| 700 | 0.5881 | 0.7551 | 0.8399 | 0.8897 | 0.9216 |

*Table 8.* Comparison of the performance of fixed and randomized trajectory points with different amount of information. "3/10" means that given a trajectory of length 10, three points are provided at indices [0, 4, 9]. Similarly, "4/10" provides points at indices [0, 3, 6, 9], and "5/10" at indices [0, 2, 4, 6, 9]. † represents the random selecting experiments.

| Info | TC@2k | TC@4k | TC@6k | TC@8k | TC@10k |
|------|-------|-------|-------|-------|--------|
| 2/10 | 0.4996 | 0.6994 | 0.8048 | 0.8667 | 0.9053 |
| 3/10 | 0.5865 | 0.7638 | 0.8498 | 0.8990 | 0.9292 |
| 4/10 | 0.6820 | 0.8305 | 0.8979 | 0.9347 | 0.9561 |
| 5/10 | 0.7362 | 0.8637 | 0.9179 | 0.9466 | 0.9633 |
| 2/10$^\dagger$ | 0.4143 | 0.6282 | 0.7402 | 0.7996 | 0.8351 |
| 3/10$^\dagger$ | 0.5364 | 0.7098 | 0.7897 | 0.8363 | 0.8663 |
| 4/10$^\dagger$ | 0.6051 | 0.7501 | 0.8269 | 0.8738 | 0.9047 |
| 5/10$^\dagger$ | 0.6817 | 0.8062 | 0.8706 | 0.9089 | 0.9336 |

## D.5. Ablation Study

Diffusion models typically involve higher computational costs, which may potentially limit their application in large-scale trajectory data scenarios. To tackle the efficiency issue, we have developed two variants that integrate DDIM sampling (ProDDIM) (Song et al., 2020) and LA (ProDDIM+Linear Attention) (Katharopoulos et al., 2020), respectively.

We carried out experiments on the WuXi dataset with k=8. As presented in Tab. 9, both variants achieve at least approximately 10× speed-up compared to ProDDPM, while only experiencing minor performance reductions. These findings indicate that our proposed model serves as a general framework. It can be effectively combined with various acceleration techniques, thereby facilitating its deployment in real-world applications and addressing the computational efficiency concerns associated with diffusion models in the context of large-scale trajectory data.

*Table 9.* Accelerated verisons of the ProDiff model (Thpt: Throughput; PPT: Processing Per Time-unit)

| Method | TC@2k | TC@4k | TC@6k | TC@8k | TC@10k | Thpt(s/sample) | PPT(sample/s) |
|--------|-------|-------|-------|-------|--------|----------------|---------------|
| ProDDPM (Ours) | 0.5752 | 0.7501 | 0.8236 | 0.8663 | 0.8945 | 77.9346 | 0.0128 |
| ProDDIM(Ours) | 0.5430 | 0.7131 | 0.7773 | 0.8303 | 0.8741 | 788.6852 | 0.0013 |
| ProDDIM+LA(Ours) | 0.5350 | 0.7197 | 0.7725 | 0.836 | 0.8785 | 768.5845 | 0.0013 |

