# OpenReview forum: "ProDiff: Prototype-Guided Diffusion for Minimal Information Trajectory Imputation"
_ICML.cc/2025/Conference — ICML 2025 poster_

### Official Review · Reviewer_xUz8 · 2025-03-12

**Overall Recommendation:** 4

**Summary:**

The paper proposes ProDiff, a prototype-guided diffusion model for trajectory imputation using only two endpoints as minimal information. It integrates prototype learning to embed human movement patterns and employs a denoising diffusion probabilistic model to reconstruct missing spatiotemporal data. A joint loss function ensures effective training. Experiments on WuXi and FourSquare datasets show that ProDiff outperforms state-of-the-art methods, with a 6.28% improvement in accuracy on FourSquare and 2.52% on WuXi. Additionally, a 0.927 correlation between generated and real trajectories suggests high fidelity.

**Claims And Evidence:**

Yes

**Essential References Not Discussed:**

They miss some reference in trajectory imputation.
For example, [1] and [2]

[1] Uncovering the Missing Pattern: Unified Framework Towards Trajectory Imputation and Prediction
[2] BCDiff: Bidirectional Consistent Diffusion for Instantaneous Trajectory Prediction
[3] Improving autonomous driving safety with pop: A framework for accurate partially observed trajectory predictions
The author should discuss them.

**Ethical Review Concerns:**

The provided GitHub repository (config/config.py, line 7) may contain identifiable author information, which may violate the double-blind review policy. The authors should ensure anonymity in all submitted materials.

**Ethical Review Flag:**

Flag this paper for an ethics review.

**Ethics Expertise Needed:**

["Research Integrity Issues (e.g., plagiarism)"]

**Experimental Designs Or Analyses:**

Yes

**Methods And Evaluation Criteria:**

Yes

**Other Comments Or Suggestions:**

none

**Other Strengths And Weaknesses:**

Strengths:
The combination of prototype learning and diffusion models is novel for trajectory imputation.

Comprehensive comparisons against state-of-the-art time-series interpolation and trajectory-specific models demonstrate consistent superiority across various trajectory window sizes.

Weaknesses
1. The paper assumes that learned prototypes can capture macro-level human movement patterns, but there is no strong theoretical justification for their optimality beyond empirical evidence.
2. They miss some reference in trajectory imputation.
For example, [1] and [2]

[1] Uncovering the Missing Pattern: Unified Framework Towards Trajectory Imputation and Prediction
[2] BCDiff: Bidirectional Consistent Diffusion for Instantaneous Trajectory Prediction
[3] Improving autonomous driving safety with pop: A framework for accurate partially observed trajectory predictions
The author should discuss them.

3. Diffusion models can be computationally expensive, and there is limited discussion on efficiency in large-scale real-time applications.

**Questions For Authors:**

none

**Relation To Broader Scientific Literature:**

Good

**Theoretical Claims:**

Yes

---

> ### Author Rebuttal · Authors · 2025-04-01
>
> **W1**: Thank you for your insightful comment. We agree the original lacked theoretical grounding and now present a concise framework supporting prototype-based modeling of macro-level human movement.
>
> ### Theoretical Justification
>
> Prototype learning combines clustering and contrastive learning. Formally, for data points $X=\{x_1,...,x_n\}$, the embedding function $f: \mathbb{R}^d \rightarrow \mathbb{R}^m$, and prototypes $\{p_1,...,p_K\}$ are optimized by:
> $$
> \min_{f,\{p_k\}}\sum_{i=1}^n\Vert f(x_i)-p_{y_i}\Vert ^2+\lambda\ell_{\mathrm{contrast}}(f(x_i),p_{y_i},\{p_k\}),
> $$
>
> **Assumptions**
> 1. **Mixture of Distributions**: Human trajectory data is drawn from a mixture of distributions, each localized on a manifold region $\mathcal{M}$ with mean $\mu_k$.
> 2. **Expressive Embeddings**: The embedding $f$ enables diverse prototypes that capture local tangents and reconstruct manifold structures via linear combinations [1].
>
> **Main Theorem**
>
> Under these assumptions, any global optimum $(f^*, \{p_k^*\})$ satisfies:
> 1. Prototypes approximate conditional expectations: $p_k^* \approx \mathbb{E}\left[f^*(x) | x \in C_k\right].$
> 2. Contrastive loss enforces prototype separation, forming diverse directional vectors:  $\langle p_i^*, p_j^* \rangle \leq \epsilon$, for $i\neq j$.
>
> **Proof Sketch**
>
> Using Pollard's consistency theorem [2], the empirical cluster centers converge to conditional expectations:
> $$
> p_k^* \approx \mathbb{E}\left[f^*(x) | x \in C_k\right]
> $$
> From InfoNCE-based contrastive loss [3], optimality conditions ensure prototype distinctiveness:
> $$
> f^*(x)^\top p_y^* - f^*(x)^\top p_k^* \ge \delta,k\neq y.
> $$
> where $ \delta>0$ . Since the clustering term already guarantees that $p_y^* \approx \mathbb{E}\left[f^*(x) \mid x\in C_y\right]$, averaging over cluster $C_y$ gives:
> $$
> \langle p_y^*, p_y^* \rangle - \langle p_y^*, p_k^* \rangle \ge \delta.
> $$
> shifting the terms leads to the observation:
> $$
> \langle p_y^*, p_k^* \rangle \le \Vert p_y^*\Vert^2 - \delta \le \epsilon, k\neq y
> $$
> which indicates that contrastive loss forces prototypes into globally distinct directions, ensuring effective representation of manifold local structures.
>
> ### Empirical Validation
>
> We validate these properties on real data:
> - Mean cosine similarity (prototypes vs. empirical means): 0.9417
> - Avg. inter-prototype angle: 84.63°
> - Avg. off-diagonal cosine similarity: 0.0915
>
> These results support the theoretical properties. Further visualizations are provided at https://anonymous.4open.science/r/ICML_rebuttal-5296/.
>
> We will incorporate these theoretical and empirical insights into the revised manuscript.
>
> **W2**：We appreciate the reviewer’s feedback and apologize for overlooking references [4-6]. We carefully reviewed [4–6] that propose unified frameworks for imputation, bidirectional diffusion, and partial observation handling,  and re-implement [4] (GC-VRNN) and [5] (BCDiff) on the WuXi dataset for direct comparison (Table below, more results at https://anonymous.4open.science/r/ICML_rebuttal-5296/).
>
> Our ProDDPM outperforms both, as it is specifically designed for human trajectory imputation, whereas GC-VRNN is designed for visual multi-agent settings and BCDiff is less aligned with our task.
>
> We will include a detailed discussion of these baselines in the revised manuscript.
> |Method|TC@2k|TC@6k|TC@10k|Time (sample/s)| Speed (s/sample)|
> |-|-|-|-|-|-|
> |GC-VRNN|0.438|0.662|0.771|7404.35|0.0001|
> |BCDiff|0.561|0.787|0.871|41.13|0.0243|
> |ProDDPM|0.575|0.824|0.895|77.93|0.0128|
> |ProDDIM|0.543|0.777|0.874|788.68|0.0013|
> |ProDDIM+LA|0.535|0.773|0.879|768.58|0.0013|
>
> **W3**: Thank you for raising this important point. we acknowledge that diffusion models generally incur higher computational costs compared to alternatives like GC-VRNN, which is faster yet significantly less accurate (GC-VRNN TC@10k=0.771 vs. ProDDPM TC@10k=0.895).
>
> To address efficiency concerns, we developed two variants that incorporate DDIM sampling (ProDDIM) and LA (ProDDIM+LA), and achieve ~10× speed-up over ProDDPM, with only minor performance reductions. These results demonstrate the practicality of ProDDIM for large-scale applications when paired with proper acceleration.
>
> We apologize for any confusion regarding anonymization and emphasize that this was an accidental mistake rather than an intentional breach of anonymity or policy violation. Upon noticing your comment, we immediately corrected the anonymized repository to eliminate your concern and prevent further misunderstandings.
>
> We hope these updates clarify the novelty and practical value of our work and respectfully invite you to reconsider the evaluation.
>
> [1] Roweis & Saul, Locally Linear Embedding. [2] Pollard, Strong Consistency of k-means. [3] Saunshi et al., Theory of Contrastive Learning. [4] GC-VRNN: Unified Framework for Trajectory Imputation. [5] BCDiff: Bidirectional Diffusion for Trajectory Prediction. [6] Improving driving safety with POP: Accurate partially observed trajectory prediction.

---

> > ### Comment · Reviewer_xUz8 · 2025-04-03
> >
> > Thanks for the rebuttal. I appreciate the efforts the authors made in the rebuttal stage. The authors provide additional theoritical analysis, experiments about two relevant method, and give two solutions for optimizing the efficiency. They address all of my concerns. Therefore, I will raise my score to accept.
> >
> > I hope the authors can revise the manuscript accordingly in the final version, including theoretical analysis, comparison with GC-VRNN and BCDiff, as well as accelerated ProDDIM/ProDDIM+LA)

---

> > > ### Author Response · Authors · 2025-04-05
> > >
> > > Thank you so much for your kind and constructive response. We are truly grateful for the time and effort you’ve spent reviewing our work and for the thoughtful suggestions you provided. Your comments greatly helped us improve the manuscript, particularly in refining the theoretical analysis, expanding the experimental comparison (with GC-VRNN and BCDiff), and optimizing the efficiency (accelerated ProDDIM and ProDDIM+LA).
> > >
> > > We will make sure to incorporate all the points you mentioned in the final version.
> > >
> > > As a small side note, we noticed that the score in the system still shows a 2 (Weak Reject). It’s possible this is just a system update delay or a small oversight, but we wanted to bring it to your attention just in case.
> > >
> > > Thanks again for your valuable feedback and support — it’s been truly helpful in improving the quality of our work.

---

### Official Review · Reviewer_NUSr · 2025-03-13

**Overall Recommendation:** 4

**Summary:**

This paper studied the task of trajectory imputation and propose ProDiff as a trajectory imputation framework that uses only two endpoints as minimal information, in order to improve previous approaches which place significant demands on data acquisition and overlook the potential of large-scale human trajectory embeddings. The experiments on FourSquare and WuXi show the proposed approach outperforms state-of-the-art methods.

**Claims And Evidence:**

The claims are supported by clear and convincing evidence.

**Essential References Not Discussed:**

N/A

**Experimental Designs Or Analyses:**

The experimental designs and ayalyses are comprehsnive and soundness for me.

**Methods And Evaluation Criteria:**

The methods and evaluation criteria make sense for the problem.

**Other Comments Or Suggestions:**

N/A

**Other Strengths And Weaknesses:**

The paper is well written. The claims in this paper are well supported, implementations details are clearly discussed and the experiments are comprehensive.

I suggest there is room for improvement in better leveraging the intermediate trajectory points when they are available. If these points are not affected by noise interference, they can deliver effective information for trajectory imputation. I agree that using the end points are able to  solve this probelm, but I also appreciate a discussion about how the proposed approach may leverge the intermediate trajectory points.

**Questions For Authors:**

N/A

**Relation To Broader Scientific Literature:**

The authors suggest that the trajectory imputation task can benefit in infectious diseases control, human behavior alanalysis, and urban planning.

**Theoretical Claims:**

The proofs for theoretical claims are correct.

---

> ### Author Rebuttal · Authors · 2025-04-01
>
> **W1：The reviewer suggested discussing how the proposed method could better leverage intermediate trajectory points when available, as they may provide valuable information for imputation beyond using only endpoints.**
>
>
>
> Thank you for this insightful comment. We fully agree that leveraging intermediate trajectory points, when available, can significantly enhance trajectory imputation performance.
>
> Although ProDiff was initially designed to operate under minimal-information conditions (using only endpoints), it can seamlessly incorporate additional intermediate points. To illustrate this capability, we conducted supplementary experiments comparing scenarios with varying amounts of intermediate trajectory points.
>
> The results (the first four rows of Tab. 1.) show performance improvements as additional fixed intermediate points are provided. Specifically, we observed significant accuracy gains as the number of known trajectory points increased from two endpoints (2/10) to five points (5/10). This clearly demonstrates the beneficial effect of integrating intermediate trajectory points. Notably, incremental performance gains became marginal beyond four points, suggesting that once essential trajectory patterns are sufficiently captured, further intermediate points provide diminishing returns.
>
>
>
> **Tab. 1. Comparison of the performance of fixed and randomized trajectory points with different amount of information.**
>
> | Fixed Points          | TC@2k  | TC@4k  | TC@6k  | TC@8k  | TC@10k |
> | --------------------- | ------ | ------ | ------ | ------ | ------ |
> | 2/10                  | 0.4996 | 0.6994 | 0.8048 | 0.8667 | 0.9053 |
> | 3/10                  | 0.5865 | 0.7638 | 0.8498 | 0.8990 | 0.9292 |
> | 4/10                  | 0.6820 | 0.8305 | 0.8979 | 0.9347 | 0.9561 |
> | 5/10                  | 0.7362 | 0.8637 | 0.9179 | 0.9466 | 0.9633 |
> | **Randomized Points** |        |        |        |        |        |
> | 2/10†                 | 0.4143 | 0.6282 | 0.7402 | 0.7996 | 0.8351 |
> | 3/10†                 | 0.5364 | 0.7098 | 0.7897 | 0.8363 | 0.8663 |
> | 4/10†                 | 0.6051 | 0.7501 | 0.8269 | 0.8738 | 0.9047 |
> | 5/10†                 | 0.6817 | 0.8062 | 0.8706 | 0.9089 | 0.9336 |
>
>
>
> Additionally, we explored scenarios where intermediate points were randomly selected each time (indicated by †), creating a more challenging environment as the points varied across instances, as shown in the last four rows of Tab. 1. This approach aimed to verify the conclusion from existing literature [1], which suggests that four randomly selected points can determine 95% of the trajectory. However, when points are selected randomly, the model’s performance diverges more noticeably from the literature’s conclusion. This discrepancy likely arises because random points disrupt the consistency of trajectory sampling, increasing the difficulty of model learning. The results highlight the potential of ProDiff in accurately imputing missing trajectory points with a sufficient number of fixed reference points, while also remains when dealing with randomly selected points, indicating an area for future improvement.
>
>
>
> [1]Unique in the crowd: The privacy bounds of human mobility.

---

### Official Review · Reviewer_8Fbd · 2025-03-14

**Overall Recommendation:** 4

**Summary:**

In this work, the authors design ProDiff, a diffusion-based model for spatial data imputation. The research direction is interesting and the problem is practical, given various noises of real-world data. ProDiff consists of two components, prototype learning and a denoising diffusion probabilistic model. With minimal information as input, the proposed method achieved good performance. Experiments on two real-world datasets demonstrate the practical effectiveness of ProDiff compared to existing methods.

**Claims And Evidence:**

claims are clear

**Essential References Not Discussed:**

References are good

**Experimental Designs Or Analyses:**

yes

**Methods And Evaluation Criteria:**

Yes, the methods make sense

**Other Comments Or Suggestions:**

no

**Other Strengths And Weaknesses:**

1.In the first paragraph, there are probably more primary sources of location data. Such as other satellite systems from Russia, CHina, Japan, etc. Also, lots of online platforms utilize IP-based locations.

2.The introduction is a bit of confusing, such as lines 64 to 68.

3.From Fig 2, it seems the diffusion base model is very similar to existing models. Maybe some clarifications about this is beneficial, e.g., is this adapted from other works or this is a typical design choice for diffusion models.

4. For experiments, is there specific pre-processing of the training data? Since the authors mention the goal is imputing trajectories with fewer constraints, the reader assumes the generation setup is more challenging than previous works.

5. I like the visualization of Fig. 5 a. However, it is hard to tell how realistic the generated results are. Actually, they look quite different. Fig 5 b,c are much better.

**Questions For Authors:**

no

**Relation To Broader Scientific Literature:**

Spatial data imputation, this direction could have broad impact in the research community such as urban science.

**Theoretical Claims:**

yes

---

> ### Author Rebuttal · Authors · 2025-04-01
>
> **W1: In the first paragraph, there are probably more primary sources of location data.**
>
> Thank you for pointing this out. We agree that our original manuscript could more comprehensively reflect the sources of trajectory data. In the revised manuscript, we will clarify this by explicitly adding other satellite systems (e.g., GLONASS from Russia, BeiDou from China, QZSS from Japan, and Galileo from Europe), and IP-based localization used by online platforms.
>
> The *updated first paragraph* will read:
>
> "Mining spatio-temporal patterns from trajectory data has broad applications in infectious disease control, human behavioral analysis, and urban planning. Such data primarily originate from Location-Based Services (LBS) using cell tower signals, satellite-based systems such as GPS, GLONASS, BeiDou, QZSS, and Galileo, as well as IP-based location methods utilized by online platforms."
>
>
> **W2: The introduction is a bit of confusing, such as lines 64 to 68.**
>
> Thank you for the comment. We agree that lines 64–68 could have been expressed more clearly. The intention was to highlight the difference between our method and prior works, but we recognize that this discussion is somewhat misplaced here. In the revised manuscript, we will refine the entire introduction to improve clarity, reorganizing the content and explicitly positioning our approach relative to existing methods in a more appropriate context.
>
>
> **W3: Clarify whether the diffusion backbone is a standard design or includes specific modifications.**
>
> Thank you for highlighting this. As noted, our diffusion backbone adopts a widely used architecture in the field which includes 1D U-Net with ResNet blocks and self-attention layers due to its strong performance in sequence modeling. This design aligns with established diffusion model practices.
>
> However, our key innovation lies in the *conditioning network and mechanism*. While prior works often use simple concatenation or shallow fusion for conditional inputs, we propose a modified Wide & Deep network that explicitly models the interaction between trajectory prototypes and conditioning signals. Specifically, We add more deep branch which is composed of multiple linear & non-linear layers that are designed to extract high-level representations from matched prototype embeddings, enabling the model to better capture trajectory patterns.  This design allows for both memorization and generalization, making the conditioning more expressive and adaptive.
>
> This integration is unique to our model and enables the diffusion process to simultaneously leverage both the base conditions and learned trajectory prototypes in a more structured and effective way. We will revise the manuscript to make these design choices and their motivations clearer.
>
>
> **W4: For experiments, is there specific pre-processing of the training data? Since the authors mention the goal is imputing trajectories with fewer constraints, the reader assumes the generation setup is more challenging than previous works.**
>
> We appreciate your question regarding preprocessing. Besides standard preprocessing (trajectory segmentation and min-max normalization), we adopt a more challenging setup: only endpoints are given as inputs, and Gaussian noise is added to intermediate points.
>
> Compared to prior methods that often use intermediate positions or extra features (e.g., speed, direction), our setting provides far less information, making the task more difficult and unconstrained.
>
> We will clarify this minimal-input design in the revision to better highlight the uniqueness and robustness of our approach.
>
>
> **W5: The realism of the generated trajectories in Fig. 5a appear less convincing compared to those in Fig. 5b and 5c.**
>
> Thank you for appreciating Fig 5b and 5c and for the thoughtful comment on Fig. 5a. To quantify these differences clearly, we computed batch-level statistics:
>
> - Mean Latitude Difference: 0.0064
> - Mean Longitude Difference: -0.0079
> - MAE Latitude: 0.0333
> - MAE Longitude: 0.0381
>
> Our batch-level statistics show that the differences between the generated and ground-truth data are minimal (~0.03–0.04°), but because we aggregate the data into grid cells with a *very fine* spatial resolution of *0.009°*, even these minute discrepancies are noticeably amplified in the visualization.
>
> Fig. 5a aims to visually illustrate alignment between generated and actual flow distributions. Figures 5a–c collectively provide complementary perspectives: spatial visualization (Fig. 5a), correlation analysis (Fig. 5b), and distributional alignment (Fig. 5c) to demonstrate that our model effectively captures realistic trajectory patterns. We will refine Fig. 5a and clarify its intent to avoid confusion in the revision.

---

> > ### Comment · Reviewer_8Fbd · 2025-04-08
> >
> > The reviewer appreciates the authors for the detailed responses and statistics of figures. I have gone through all the responses and the paper again and will raise my rating.  Hope to see the revised version soon.

---

> > > ### Author Response · Authors · 2025-04-08
> > >
> > > We sincerely thank the reviewer for the careful reading of our manuscript and the thoughtful comments. We greatly appreciate your positive feedback and are glad that our responses and additional analyses were helpful. Your suggestions are invaluable and will undoubtedly contribute to improving the quality of our work.
> > >
> > > We will incorporate the relevant information into the revised manuscript accordingly. Thank you again for your time and constructive review.

---

### Decision · Program_Chairs · 2025-05-01

**Decision:**

Accept (poster)

**Comment:**

All reviewers are positive about the paper. The combination of prototype learning and diffusion models is well received as a valid approach to address the trajectory imputation task. The authors provide comprehensive comparisons against state-of-the-art time-series interpolation and trajectory-specific models. They also did a good job during the rebuttal in addressing the reviewers’ questions.
The ACs echo the reviewers' comments and recommend including the added theoretical analysis and new experiments (e.g., comparisons with GC-VRNN, BCDiff, and ProDDIM/ProDDIM+LA) and the final camera-ready version.